# Analysis of Factors Affecting Post-Stroke Fatigue: An Observational, Cross-Sectional, Retrospective Chart Review Study

**DOI:** 10.3390/healthcare9111586

**Published:** 2021-11-19

**Authors:** Seungwon Kwon, Chul Jin, Seung-Yeon Cho, Seong-Uk Park, Woo-Sang Jung, Sang-Kwan Moon, Jung-Mi Park, Chang-Nam Ko, Ki-Ho Cho

**Affiliations:** Department of Cardiology and Neurology, College of Korean Medicine, Kyung Hee University, Seoul 02447, Korea; yahaly@naver.com (C.J.); sy.cho@khu.ac.kr (S.-Y.C.); seonguk.kr@gmail.com (S.-U.P.); wsjung@khu.ac.kr (W.-S.J.); skmoon@khu.ac.kr (S.-K.M.); pajama@khu.ac.kr (J.-M.P.); kcn202@khu.ac.kr (C.-N.K.); kihocho58@gmail.com (K.-H.C.)

**Keywords:** post-stroke fatigue, depression, cognitive dysfunction, platelet to lymphocyte ratio, retrospective chart review study

## Abstract

Post-stroke fatigue (PSF) is one of the most common emotional and mood disorders in stroke survivors. Several studies have suggested associations between PSF and various factors. However, they describe conflicting results. Therefore, this study aimed to evaluate the factors affecting PSF. We retrospectively reviewed the medical records of 178 hospitalized stroke patients. The collected data were compared between the PSF and control groups. To evaluate the association between factors and PSF, regression analysis was conducted. A total of 96 patients (53.9%) were assigned to the PSF group, and 82 patients were assigned to the control group. Age, neurological deficits, cognitive dysfunction, degree of depression, hs-CRP, and ESR differed significantly between the two groups. For both types of stroke, multiple linear regression analyses showed that degree of depression and degree of inflammation were significantly associated with PSF. Through subgroup analysis, multiple linear regression analyses showed that the degree of depression in ischemic and hemorrhagic stroke and the platelet-to-lymphocyte ratio in hemorrhagic stroke had a significant association with PSF. In conclusion, post-stroke depression and degree of inflammation could be clinically significant predictors of PSF in all types of stroke patients. However, larger, prospective studies are required to obtain more concrete results.

## 1. Introduction

Despite the improvements made in the survival rate of patients with stroke [1], these patients still experience physical disability, as well as cognitive, emotional, and behavioral disorders as sequelae [2], all of which significantly reduce the quality of life of survivors [2]. In the past, stroke treatment strategies mainly focused on physical conditions such as motor or sensory disorders. However, with the recent improvements in the quality of life of survivors, the importance of managing emotional and mood disorders, which are non-physical disorders, has also been highlighted [3].

Post-stroke emotional disorders encompass depression, anxiety, emotion control disorder, anger control disorder, and fatigue that occurs after a stroke [3]. Post-stroke fatigue (PSF) is one of the most common emotional disorders, and the previous studies suggested that 16%–85% of stroke patients experience PSF [4,5,6,7]. PSF is known to adversely affect the overall prognosis of patients with stroke. In one review, PSF was shown to lead to physical deconditioning and reduced self-efficacy in physical performance, poor participation, and outcomes in rehabilitation programs, reduced social participation, poor quality of life, functional limitations, and increased mortality [7,8].

Thus, establishment of an effective treatment strategy for PSF is essential for efficient recovery of stroke survivors. However, a clear explanation of the pathophysiology of PSF, which will serve as the basis for establishing a treatment strategy, is still lacking. The relationships among age, sex, type of stroke, severity of neurological deficits, accompanying symptoms (e.g., depression, sleep disturbance, etc.), autonomic nervous system abnormalities, inflammation, and neuroendocrine dysregulation have been highlighted in a recent study [7]. However, the results of some studies are contradictory, highlighting the need for additional research. Moreover, since the pathophysiology and treatment application for each type of stroke are different [9], the factors contributing to the development of fatigue symptoms may differ even after the same stroke, and information on these seems to be lacking.

Thus, in the present study, to establish an effective treatment strategy for PSF, a retrospective chart review was used along with analyses of the known physical and mental factors that could contribute to fatigue in stroke patients, as well as the effects of inflammatory biomarkers such as neutrophil-to-lymphocyte ratio (NLR), monocyte-to-lymphocyte ratio (MLR), and platelet-to-lymphocyte ratio (PLR). Factors contributing to fatigue in each stroke type (cerebral infarction and cerebral hemorrhage) were further analyzed.

## 2. Methods

### 2.1. Study Participants

#### 2.1.1. Selection Criteria

The medical records of patients meeting the following criteria were selected for the study: (i) patients aged 19 years or older who were hospitalized under the Korean standard classification of disease and causes of death codes I63 (cerebral infarction) or I61 (intracerebral hemorrhage) at Kyung Hee University Korean Medicine Hospital, Stroke and Brain Disease Center from 1 March, 2019, to 29 February, 2020; and (ii) patients who answered questionnaires for the Fatigue Assessment Scale (FAS), Fatigue Severity Scale (FSS), and subjective fatigue, and underwent blood tests at admission. 

#### 2.1.2. Exclusion Criteria

Medical records of patients who met the following criteria were excluded from the study: (i) patients with degenerative brain diseases (e.g., Parkinson disease, Alzheimer’s disease, etc.), brain tumors, and other brain diseases such as traumatic brain injury; and (ii) patients with pre-existing disabilities due to other neurological or myologic disorders before the onset of cerebrovascular disease.

### 2.2. Methods

This study was approved by the Institutional Review Board of Kyung Hee University Korean Medicine Hospital (KOMCIRB 2020-12-003). The study was conducted using a retrospective chart review method through a medical records survey. The protocol of this study was registered in the Clinical Research Information Service (KCT0005992). The consent of individual participants was not required for this study, as it was conducted using data from the medical records that did not include participants’ personally identifiable information. The exemption from the process of acquiring consent was approved by the institutional review board. In addition, this study was conducted in compliance with the clinical research guidelines, such as the Declaration of Helsinki.

The participants selected after application of the selection and exclusion criteria were classified into post-stroke fatigue (PSF) and control (non-post-stroke fatigue [non-PSF]) groups. Participants who showed two or more of the following findings were classified into the PSF group: (i) subjective symptoms of fatigue; (ii) total FAS score of 24 or higher [10]; and (iii) total FSS score of 4 or higher [11]. Subsequently, the following data were collected for the patients assigned to each group: (i) demographic characteristics; (ii) stroke-related characteristics; and (iii) laboratory test results.

### 2.3. Observation Items

The following items were collected through medical record inquiry.

#### 2.3.1. Demographic Characteristics

(1)Age (years);(2)Sex (male, female);(3)Body mass index (kg/m^2^);(4)Education level (years).

#### 2.3.2. Stroke-Related Characteristics

(1)Stroke hospitalization duration and disease duration;(2)Stroke type classification: ischemic stroke and hemorrhagic stroke;(3)History of stroke surgery;(4)Family history of stroke;(5)Degree of neurological impairment: National Institutes of Health Stroke Scale (NIHSS) score;(6)Cognitive function: Korean version of the Mini-Mental State Examination (MMSE-K) score;(7)Depression: Patient Health Questionnaire (PHQ-9) score;(8)History of stroke and various risk factors.

At the time of hospitalization, the medical records were evaluated to determine information related to stroke, hypertension, dyslipidemia, diabetes, heart disease, and a history of cancer.

#### 2.3.3. Lab Test Results

At the time of admission, the results of the laboratory tests performed on the same day or the next day were collected.

(1)Biochemical, endocrine, and lipid tests

Total protein, albumin, total bilirubin, blood urea nitrogen (BUN), creatinine, aspartate aminotransferase (AST), alanine aminotransaminase (ALT), gamma-glutamyl transferase (γ-GT), thyroid stimulating hormone (TSH), hemoglobin A1c (HbA1c), homocysteine, total cholesterol, triglyceride, low-density lipoprotein (LDL), high-density lipoprotein (HDL), apolipoprotein A1, apolipoprotein B, total lipid, phospholipid, and high-sensitivity C-reactive protein (hs-CRP) levels were investigated.

(2)Complete blood count and inflammatory marker tests

White blood cell (WBC) count, red blood cell (RBC) count, hemoglobin level, platelet count, erythrocyte sedimentation rate (ESR), and lymphocyte, monocyte, eosinophil, basophil, and neutrophil counts were investigated. On the basis of these results, NLR, MLR, and PLR, which have attracted attention as novel inflammatory markers, were calculated.

### 2.4. Statistical Analysis 

To investigate the differences in characteristics according to the presence or absence of PSF, *t*-tests (cases with normal distribution) or Mann–Whitney U test (cases that do not show a normal distribution) were performed for continuous variables, and chi-squared tests were performed for categorical variables. In addition, multiple linear regression tests were performed using the backward method to identify the factors affecting PSF. In the regression analysis, the independent variables were defined as FAS and FSS scores, and the dependent variables included variables with a significance probability of less than 0.10 in the comparison between the groups. All statistical analyses were performed using the IBM SPSS Statistics ver. 25 (IBM Co., Armonk, NY, USA).

## 3. Results

Medical record inquiry identified a total of 302 patients who received inpatient treatment under the main diagnosis codes of I63 or I61 at the Stroke and Brain Disease Center at Kyung Hee University Korean Medicine Hospital between 1 March 2019 and 29 February 2020. Among them, 124 patients who did not meet the selection criteria were excluded. Finally, the medical records of 178 patients (144 patients with ischemic stroke and 34 patients with hemorrhagic stroke) were included in the analysis; of these, 96 patients were allocated to the PSF group and 82 were assigned to the control group on the basis of the group assignment criteria.

### 3.1. Comparison of Demographic Characteristics

The PSF and control groups showed no significant differences in terms of sex ratio, body mass index, and education period. The average age was 69.0 ± 10.5 years in the PSF group, and 65.0 ± 12.9 years in the control group, indicating a statistically significant difference (*p* = 0.027) between the two groups. In the analysis of ischemic stroke patients, the mean age showed a statistically significant difference between the two groups (PSF group vs. control group: 71.0 ± 9.9 vs 66.8 ± 11.7 years, *p* = 0.018). However, among patients with hemorrhagic stroke, the two groups showed no significant differences in terms of age, sex ratio, body mass index, and education period (Table 1).

### 3.2. Comparison of Stroke-Related Characteristics

The PSF and control groups showed no significant differences in terms of length of hospital stay, stroke prevalence, stroke-related surgery rate, and family history of stroke. The degree of neurological deficit assessed by the NIHSS score, as well as the history of stroke and stroke-related risk factors, were not significantly different between the two groups. On the other hand, the neurological deficits evaluated by NIHSS score showed a statistically significant difference between the groups with a higher score in the PSF group than in the control group (PSF group vs. control group: 5.65 ± 4.5 vs. 4.51 ± 4.1, *p* = 0.038). The cognitive function evaluated by the MMSE-K score showed a statistically significant difference between the groups, with a lower score in the PSF group than in the control group (PSF group vs. control group: 23.33 ± 6.9 vs. 25.25 ± 3.9, *p* = 0.006). The degree of depression after stroke, evaluated by the PHQ-9 score, also showed a statistically significant difference, with the PSF group showing a more severe pattern than the control group (PSF group vs. control group: 11.10 ± 6.0 vs. 3.83 ± 3.2, *p* < 0.001; Table 2).

In the analysis of patients with ischemic stroke, the two groups showed a significant difference in the degree of cognitive function evaluated by the MMSE-K score (PSF group vs. control group: 23.38 ± 6.9 vs. 24.92 ± 4.0, *p* = 0.014), and the degree of post-stroke depression evaluated by the PHQ-9 score (PSF group vs. control group: 1.130 ± 6.4 vs. 3.65 ± 3.1, *p* < 0.001), and the degree of cognitive dysfunction and depression were more severe in the PSF group. Among patients with hemorrhagic stroke, the PSF group showed significantly higher degree of severe depression (PSF group vs. control group: 10.38 ± 4.7 vs. 4.77 ± 3.5, *p* = 0.001), and a higher prevalence of hypertension (PSF group vs. control group: 66.7% vs. 30.8%, *p* = 0.046) (Table 2).

### 3.3. Comparison of Laboratory Evaluation Results

The two groups showed no significant differences in the biochemical, endocrine, and lipid test items evaluated at the time of hospitalization. A similar trend was observed in the analysis of patients with ischemic stroke and hemorrhagic stroke. In addition, the two groups showed no significant differences in the results of general hematological analyses. On the other hand, the hs-CRP and ESR, inflammatory markers, were significantly higher in the PSF group (PSF group vs. control group: 0.77 ± 1.77 vs. 0.52 ± 1.35, *p* = 0.011; PSF group vs. control group: 30.07 ± 21.40 vs. 23.86 ± 23.00, *p* = 0.006). This trend was also confirmed in ESR values of ischemic stroke patients (PSF group vs. control group: 30.51 ± 22.55 vs. 24.32 ± 24.28, *p* = 0.018), but the ESR values did not differ significantly between the PSF and control groups in hemorrhagic stroke patients (Table 3 and Table 4).

### 3.4. Multivariate Analysis of Factors Affecting Post-Stroke Fatigue

A multiple linear regression test was performed to investigate the independent influence of each factor on the difference between the PSF and the control groups, confirmed through group comparison. For all stroke patients, age, stroke surgery history, NIHSS scores, PHQ-9 score, MMSE-K score, hs-CRP, and ESR were adjusted as confounding factors. A significant positive association was found for the PHQ-9 score in FAS and FSS scores (*p* < 0.001), and ESR in FAS scores (*p* = 0.047) (Adjusted R^2^ = 0.452, Root Mean Square Error (MSE) = 7.819, Table 5).

Age, hospitalization duration, PHQ-9 scores, MMSE-K scores, hs-CRP, ESR, and AST were adjusted as confounding factors in patients with ischemic stroke. A significant positive association was confirmed for the PHQ-9 score in FAS and FSS scores (*p* < 0.001, adjusted R^2^ = 0.490, Root Mean Square Error (MSE) = 7.541). In hemorrhagic stroke patients, after adjusting for PHQ-9 scores, hypertension prevalence, PLR, hs-CRP, and albumin as confounding factors, a significant positive association was found with PHQ-9 score in FAS and FSS scores (*p* = 0.001), as well as PLR in FAS and FSS scores (*p* = 0.007, 0.048 in each) (Adjusted R^2^ = 0.490, Root Mean Square Error (MSE) = 7.995, Table 5).

## 4. Discussion

This study involved a retrospective medical record analysis to confirm the characteristics of patients with PSF. Demographic characteristics, stroke-related characteristics, and laboratory test results were collected from patients hospitalized for treatment after stroke for one year at a medical institution. After analyzing the differences in characteristics between the PSF and control groups, the stroke-related factors that had a significant effect on PSF were investigated. A total of 178 stroke patients who met the selection criteria were analyzed. The number of patients assigned to the PSF group was 96 (53.9%). This finding was consistent with the 57% prevalence of PSF reported in a previous study [12], which is thought to indicate some degree of homogeneity with the participant characteristics in the previous study.

The multiple linear regression analysis confirmed that the degree of depression evaluated by the PHQ-9 score and the degree of inflammation evaluated by the ESR were significantly related to fatigue in all stroke patients. Among these, the relationship between depression and PSF requires attention. The association between PSF and depression [12,13,14,15] has been suggested several times in previous studies, and it was also confirmed in this study. These findings indicate that depression plays important roles in the development of PSF. The PHQ-9 is the most widely used screening tool for depression, and the higher the score, the more severe the depression [16]. A total score of nine or more was defined as depression [16]. In this study, the average PHQ-9 score of the PSF group was 11.10 ± 6.0, a value that can be considered to indicate a post-stroke depressive state. This trend was similar in both patients with ischemic and hemorrhagic stroke.

In this study, we also tried to determine the relationship between various physical factors and the occurrence of PSF, in addition to psychological factors such as depression. To this end, we analyzed the results of various laboratory tests such as biochemical, endocrine, lipid, general hematological, and inflammatory marker tests. The results confirmed that hs-CRP, ESR, and PLR, which are representative inflammatory markers, show significant positive association with the severity of fatigue in all types of stroke patients and hemorrhagic stroke patients. Among them, we initially focused on hs-CRP. Hs-CRP has been used to predict the prognosis of cardiovascular [17] and cerebrovascular disorders [18]. A previous study also suggested that PSF 6 months after a stroke was linked to elevated plasma hs-CRP levels upon admission, and this outcome is in line with the findings of the present study [19]. Another inflammatory marker, PLR, has been recently gaining attention as an indicator that reflects the inflammatory state of the body, and is also used as a prognostic predictor in cancer patients [20,21,22], cardiovascular disease patients [23,24], stroke patients [25], and coronavirus disease-19 (COVID-19) patients. It is also attracting attention as a predictor of cytokine storm in patients infected with COVID-19 in the context of a pandemic [26,27]. An existing review [28] has already introduced evidence related to the association between PSF and various inflammatory cytokines. That review [28] raised the possibility that stroke-related inflammatory processes could affect the formation of fatigue symptoms after stroke; however, further studies are needed. The results of this study, therefore, are meaningful in that they confirmed that the occurrence of fatigue in stroke patients and stroke-related inflammatory processes have a significant relationship.

In addition, to explain the relationship between stroke-related inflammatory processes and PSF, various interleukins and inflammatory cytokines, such as tumor necrosis factor-α (TNF-α), which are difficult to use in daily clinical settings, were used as indicators. On the other hand, in this study, the novel inflammatory markers PLR, MLR, and NLR, which can be calculated easily by performing general hematology tests and differential counts that are frequently used in general clinical settings, were utilized. Therefore, the results of this study can be helpful in determining whether to apply stroke-related inflammation-associated measures when establishing a treatment strategy for patients complaining of PSF in front-line clinical settings.

Furthermore, in this study, patients with ischemic stroke and those with hemorrhagic stroke were analyzed separately. As a result, only depression showed a significant association in ischemic stroke patients, but depression and stroke-related inflammation were confirmed to be significantly associated with hemorrhagic stroke. Although the number of patients with hemorrhagic stroke was relatively small and no definite conclusions can be drawn yet, the results suggest that not only do the two types of strokes have different mechanisms of disease development [9]; they may also require different treatment strategies for fatigue symptoms.

This study had several limitations. First, since only inpatients of a single medical institution were investigated, the overall severity of stroke is likely to be higher than that among outpatients. Therefore, in future studies, it may be necessary to reconfirm the results of this study by expanding the survey group to outpatients. Second, the PHQ-9 scores used to evaluate depression show limitations in yielding specific and detailed information when they are widely used as screening tools. Therefore, in future studies, it is necessary to use the Beck Depression Inventory [29] or the Hamilton Depression Rating Scale [30] to evaluate depressive symptoms. Lastly, since this study was conducted as an observational cross-sectional study as well as a retrospective medical record analysis study, there are limitations in causal inference. In the future, causality inference through prospective studies will need to be conducted.

Despite these limitations, this study is meaningful in that it extensively analyzed various physical and mental factors that affect the occurrence of PSF. The results showed that it was possible to reconfirm the importance of previously known mental factors such as depression, and to confirm the relationship between various cytokines and PSF through inflammatory markers such as hs-CRP, ESR, and PLR, which are convenient to use in clinical settings. In the future, studies that can supplement the limitations of this study should be conducted to identify more confirmatory factors for PSF. These efforts are expected to contribute to the development of new treatment strategies for PSF.

## Figures and Tables

**Table 1 healthcare-09-01586-t001:** Comparison of demographic characteristics between the PSF and non-PSF groups.

	All Types of Stroke			Ischemic Stroke			Hemorrhagic Stroke		
	PSF (*n* = 96)	Non-PSF (*n* = 82)	*p*-Value ^a^	PSF (*n* = 75)	Non-PSF (*n* = 69)	*p*-Value ^a^	PSF (*n* = 21)	Non-PSF (*n* = 13)	*p*-Value ^a^
Age, year	69.0 ± 10.5	65.0 ± 12.9	0.027 ^b^	71.0 ± 9.9	66.8 ± 11.7	0.018 ^b^	61.9 ± 9.7	55.5 ± 15.3	0.140
Sex, n (%)									
Male	47 (49.0)	48 (58.5)	0.130	37 (49.3)	41 (59.4)	0.148	10 (47.6)	7 (53.8)	0.500
Female	49 (51.0)	34 (41.5)		38 (50.7)	28 (40.6)		11 (52.4)	6 (46.2)	
BMI, kg/m^2^	24.12 ± 3.31	24.33 ± 3.66	0.689	24.24 ± 3.55	24.53 ± 3.44	0.495 ^b^	23.68 ± 2.28	23.23 ± 4.69	0.748
Education, years	7.8 ± 5.5	8.7 ± 5.3	0.611 ^b^	7.3 ± 5.3	8.1 ± 5.2	0.749 ^b^	9.6 ± 5.9	11.5 ± 5.3	0.400 ^b^

Values are mean ± SD or number (%); PSF group: Patients with post-stroke fatigue; non-PSF group: patients without post-stroke fatigue; BMI: body mass index. ^a^
*t*-test or Mann–Whitney U test for continuous variables and Chi-squared test for categorical variables were used. ^b^ The *p*-value for this variable was calculated using the Mann–Whitney U test.

**Table 2 healthcare-09-01586-t002:** Comparison of disease characteristics between the PSF and non-PSF groups.

	All Types of Stroke			Ischemic Stroke			Hemorrhagic Stroke		
	PSF (*n* = 96)	Non-PSF (*n* = 82)	*p*-Value ^a^	PSF (*n* = 75)	Non-PSF (*n* = 69)	*p*-Value ^a^	PSF (*n* = 21)	Non-PSF (*n* = 13)	*p*-Value ^a^
Hospitalization duration, days	33.0 ± 32.0	27.5 ± 26.6	0.113 ^b^	31.4 ± 26.9	25.6 ± 26.0	0.059 ^b^	38.8 ± 46.1	37.5 ± 28.7	0.727 ^b^
Disease duration, months	1.2 ± 0.7	1.7 ± 3.2	0.814 ^b^	1.1 ± 0.5	1.5 ± 2.3	0.544 ^b^	1.5 ± 1.2	3.1 ± 6.0	0.484 ^b^
Stroke surgery history, n (%)	14 (14.6)	6 (7.3)	0.097	5 (6.7)	3 (4.3)	0.407	9 (42.9)	3 (23.1)	0.212
Family history, n (%)	21 (21.9)	19 (23.2)	0.488	15 (20.0)	18 (26.1)	0.251	6 (28.6)	1 (7.7)	0.153
NIHSS score	5.65 ± 4.5	4.51 ± 4.1	0.038 ^b^	5.20 ± 4.1	4.54 ± 4.3	0.162 ^b^	7.24 ± 5.5	4.33 ± 3.1	0.131 ^b^
MMSE-K score	23.33 ± 6.9	25.25 ± 3.9	0.006 ^b^	23.38 ± 6.9	24.92 ± 4.0	0.014 ^b^	23.20 ± 6.7	26.92 ± 2.7	0.169 ^b^
PHQ-9 score	11.10 ± 6.0	3.83 ± 3.2	<0.001	11.30 ± 6.4	3.65 ± 3.1	<0.001	10.38 ± 4.7	4.77 ± 3.5	0.001
Medical history, n (%)									
Stroke	21 (21.9)	19 (23.2)	0.488	15 (20.0)	18 (26.1)	0.251	6 (28.6)	1 (7.7)	0.153
Hypertension	62 (64.6)	55 (64.7)	0.425	48 (64.0)	51 (73.9)	0.135	14 (66.7)	4 (30.8)	0.046
Dyslipidemia	33 (34.4)	31 (37.8)	0.375	31 (41.3)	29 (42.0)	0.534	2 (9.5)	2 (15.4)	0.498
Diabetes mellitus	25 (26.0)	26 (31.7)	0.252	22 (29.3)	24 (34.8)	0.301	3 (14.3)	2 (15.4)	0.647
Heart disease	22 (22.9)	14 (17.1)	0.218	19 (25.3)	14 (20.3)	0.302	3 (14.3)	0 (0.0)	0.222
Cancer	5 (5.2)	5 (6.1)	0.524	5 (6.7)	5 (7.2)	0.574	0 (0.0)	0 (0.0)	1.000

Values are mean ± SD or number (%). PSF group: patients with post-stroke fatigue; non-PSF group: patients without post-stroke fatigue NIHSS: National Institutes of Health Stroke Scale; MMSE-K: Korean version of the Mini-Mental State Examination; PHQ-9, Patient Health Questionnaire. ^a^
*t*-test or Mann–Whitney U test for continuous variables and Chi-squared test for categorical variables were used. ^b^ the *p*-value for this variable was calculated using the Mann–Whitney U test.

**Table 3 healthcare-09-01586-t003:** Comparison of biochemical data between the PSF and non-PSF groups.

	**All Types of Stroke**			**Ischemic Stroke**			**Hemorrhagic Stroke**		
	**PSF** **(*n* = 96)**	**Non-PSF** **(*n* = 82)**	***p*-Value ^a^**	**PSF** **(*n* = 75)**	**Non-PSF** **(*n* = 69)**	***p*-Value ^a^**	**PSF** **(*n* = 21)**	**Non-PSF** **(*n* = 13)**	***p*-Value ^a^**
Total protein, g/dL	6.97 ± 0.64	6.93 ± 0.89	0.638 ^b^	6.96 ± 0.60	6.88 ± 0.94	0.751 ^b^	7.00 ± 0.78	7.19 ± 0.45	0.435
Albumin, g/dL	3.99 ± 0.39	4.62 ± 4.36	0.164	3.97 ± 0.37	4.69 ± 4.76	0.208	4.04 ± 0.43	4.30 ± 0.42	0.098
Total bilirubin, mg/dL	1.52 ± 8.49	0.66 ± 0.26	0.441 ^b^	1.79 ± 9.66	0.67 ± 0.27	0.627 ^b^	0.58 ± 0.43	0.61 ± 0.17	0.583
BUN, mg/dL	18.19 ± 8.53	16.90 ± 6.57	0.391 ^b^	18.99 ± 8.47	16.68 ± 5.94	0.112 ^b^	15.33 ± 8.34	18.08 ± 9.48	0.462 ^b^
Creatinine, mg/dL	0.85 ± 0.35	0.92 ± 0.81	0.902 ^b^	0.89 ± 0.37	0.87 ± 0.51	0.636 ^b^	0.71 ± 0.25	1.22 ± 1.67	0.701 ^b^
AST, U/L	35.00 ± 43.63	26.56 ± 12.26	0.192 ^b^	35.19 ± 41.41	26.96 ± 12.72	0.065 ^b^	34.33 ± 51.91	24.46 ± 9.60	0.505
ALT, U/L	35.17 ± 58.41	25.17 ± 16.00	0.529 ^b^	36.39 ± 62.03	25.57 ± 16.58	0.315 ^b^	30.81 ± 44.11	23.08 ± 12.80	0.780 ^b^
γ-GT, U/L	31.42 ± 22.57	37.24 ± 45.74	0.757 ^b^	33.93 ± 24.48	38.33 ± 48.23	0.420 ^b^	22.43 ± 96	31.46 ± 30.01	0.600 ^b^
TSH, mIU/L	2.49 ± 5.19	2.29 ± 2.10	0.350 ^b^	2.45 ± 5.63	2.21 ± 2.00	0.216 ^b^	2.61 ± 3.10	2.78 ± 2.68	0.933 ^b^
HbA1c, %	6.77 ± 5.25	6.12 ± 1.04	0.590 ^b^	6.32 ± 1.27	6.51 ± 1.07	0.844 ^b^	8.36 ± 11.01	5.58 ± 0.63	0.136 ^b^
Homocysteine, μmol/L	11.65 ± 7.44	11.31 ± 5.30	0.983 ^b^	10.83 ± 4.54	11.32 ± 4.56	0.557	14.94 ± 13.84	11.26 ± 8.97	0.196 ^b^
Total Cholesterol, mg/dL	148.52 ± 45.64	154.69 ± 54.59	0.745 ^b^	137.52 ± 40.80	152.42 ± 56.47	0.249 ^b^	185.71 ± 42.07	167.17 ± 42.63	0.234
Triglyceride, mg/dL	120.97 ± 58.54	125.49 ± 6.03	0.796 ^b^	119.10 ± 59.96	127.46 ± 65.79	0.591 ^b^	127.19 ± 54.44	115.62 ± 47.79	0.484 ^b^
LDL-Cholesterol, mg/dL	88.31 ± 30.89	89.27 ± 33.34	0.896 ^b^	80.27 ± 26.69	87.06 ± 32.98	0.398 ^b^	115.10 ± 29.26	100.46 ± 34.23	0.193
HDL-Cholesterol, mg/dL	43.79 ± 10.71	47.48 ± 20.5	0.316 ^b^	43.31 ± 10.22	48.23 ± 22.06	0.224 ^b^	45.43 ± 12.37	43.69 ± 10.17	0.674
hs-CRP, mg/dL	0.77 ± 1.77	0.52 ± 1.35	0.011 ^b^	0.68 ± 1.79	0.58 ± 1.47	0.056 ^b^	1.17 ± 1.73	0.21 ± 0.31	0.053

Values are mean ± SD. PSF group: patients with post-stroke fatigue; non-PSF group: patients without post-stroke fatigue BUN: blood urea nitrogen; AST: aspartate aminotransferase; ALT: alanine transaminase; γ-GT: gamma-glutamyl transferase; TSH: thyroid stimulating hormone; HbA1c: glycated hemoglobin; LDL: low-density lipoprotein; HDL: high-density lipoprotein; CRP: c-reactive protein. ^a^
*t*-test or Mann–Whitney U test for continuous variables and Chi-squared test for categorical variables were used. ^b^ the *p*-value for this variable was calculated using the Mann–Whitney U test.

**Table 4 healthcare-09-01586-t004:** Comparison of hematologic data between the PSF and non-PSF groups.

	All Types of Stroke			Ischemic Stroke			Hemorrhagic Stroke		
	PSF (*n* = 96)	Non-PSF (*n* = 82)	*p*-Value ^a^	PSF (*n* = 75)	Non-PSF (*n* = 69)	*p*-Value ^a^	PSF (*n* = 21)	Non-PSF (*n* = 13)	*p*-Value ^a^
WBC, 10^3^/μL	7.05 ± 2.43	6.65 ± 2.52	0.229 ^b^	7.23 ± 2.58	6.87 ± 2.62	0.400 ^b^	6.42 ± 1.73	5.48 ± 1.46	0.115
RBC, 10^6^/μL	4.29 ± 0.53	4.34 ± 0.53	0.540	4.29 ± 0.56	4.36 ± 0.53	0.452	4.31 ± 0.43	4.26 ± 0.55	0.781
Hemoglobin, g/dL	13.27 ± 1.72	13.51 ± 1.73	0.364	13.31 ± 1.79	13.51 ± 1.74	0.505	13.12 ± 1.47	13.48 ± 1.69	0.510
Platelet, 10^3^/μL	249.93 ± 90.99	237.4 ± 84.00	0.702 ^b^	245.29 ± 96.29	237.02 ± 89.01	0.827 ^b^	266.48 ± 68.19	239.77 ± 52.17	0.148 ^b^
ESR, mm/hr	30.07 ± 21.40	23.86 ± 23.00	0.006 ^b^	30.51 ± 22.55	24.32 ± 24.28	0.018 ^b^	28.52 ± 17.13	21.25 ± 13.92	0.220
Segment of Lymphocyte, %	27.30 ± 11.03	31.29 ± 25.80	0.197 ^b^	26.82 ± 10.92	28.23 ± 10.22	0.315 ^b^	29.00 ± 11.53	47.56 ± 59.68	0.172
Monocyte, %	5.86 ± 1.40	5.99 ± 1.49	0.457 ^b^	5.98 ± 1.46	6.01 ± 1.51	0.692 ^b^	5.45 ± 1.08	5.88 ± 1.46	0.330
Eosinophil, %	3.04 ± 2.30	2.51 ± 1.97	0.119 ^b^	2.95 ± 2.08	2.52 ± 2.02	0.174 ^b^	3.36 ± 3.02	2.44 ± 1.74	0.441 ^b^
Basophil, %	0.51 ± 0.38	0.57 ± 0.56	0.396 ^b^	0.51 ± 0.40	0.58 ± 0.60	0.291 ^b^	0.52 ± 0.30	0.48 ± 0.22	0.686
Neutrophil, %	62.33 ± 10.39	61.30 ± 8.11	0.460	63.04 ± 9.79	61.77 ± 8.48	0.408	59.81 ± 12.22	58.85 ± 5.39	0.755
NLR	2.84 ± 1.85	2.41 ± 1.11	0.280 ^b^	2.87 ± 1.79	2.51 ± 1.13	0.360 ^b^	2.74 ± 2.07	1.88 ± 0.83	0.506 ^b^
MLR	0.25 ± 0.11	0.23 ± 0.11	0.329 ^b^	0.25 ± 0.11	0.24 ± 0.11	0.375 ^b^	0.23 ± 0.13	019 ± 0.10	0.441 ^b^
PLR	10.79 ± 6.76	8.94 ± 4.63	0.123 ^b^	10.72 ± 6.99	9.25 ± 4.86	0.371 ^b^	11.05 ± 6.00	7.26 ± 2.73	0.082 ^b^

Values are mean ± SD. PSF group: Patients with post-stroke fatigue; non-PSF group: patients without post-stroke Fatigue; WBC: white blood cell; RBC: red blood cell; ESR: erythrocyte sedimentation rate; NLR: neutrophil-to-lymphocyte ratio; MLR: monocyte-to-lymphocyte ratio; PLR: platelet-to-lymphocyte ratio. ^a^
*t*-test or Mann–Whitney U test for continuous variables and Chi-squared test for categorical variables were used. ^b^ the *p*-value for this variable was calculated using the Mann–Whitney U test.

**Table 5 healthcare-09-01586-t005:** Results of multiple linear regression model for post-stroke fatigue.

	All Types of Stroke			Ischemic Stroke			Hemorrhagic Stroke		
Factors	Estimate	Standard Error	*p*-Value ^a^	Estimate	Standard Error	*p*-Value ^a^	Estimate	Standard Error	*p*-Value ^a^
(a) FAS scores									
ESR	0.065	0.032	0.047						
PLR							0.829	0.279	0.007
AST				0.035	0.021	0.094			
PHQ-9	1.103	0.117	<0.001	1.149	0.117	<0.001	1.143	0.296	0.001
(b) FSS scores									
Hypertension							−10.227	5.438	0.073
PLR							0.955	0.456	0.048
PHQ-9	1.868	0.1941	<0.001	1.837	0.201	<0.001	1.884	0.506	0.001

*p*-values were evaluated using the multiple linear regression test; ESR: erythrocyte sedimentation rate; PLR: Platelet to Lymphocyte Ratio; AST: aspartate aminotransferase; PHQ-9: Patient Health Questionnaire-9. ^a^*t*-test or Mann–Whitney U test for continuous variables and Chi-squared test for categorical variables were used.

## Data Availability

The data presented in this study are available on request from the corresponding author. The data are not publicly available, because the data of present study was obtained from medical records.

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
