# Peer review of "Analysis of Factors Affecting Post-Stroke Fatigue: An Observational, Cross-Sectional, Retrospective Chart Review Study"

_healthcare, 2021, doi:10.3390/healthcare9111586_

Round 1

Reviewer 1 Report

The authors in a retrospective chart study investigated fatigue after ischemic and hemorraghic stroke in 178 pts in one Korean institution. 54% of patients were classified as in fatigue, the rest not. Clinical and lab variables were compared between groups (and subgroups), and a few factors proved significant. Linear regression analysis proved significant effects of cognitive decline and depression as well as a biochemical parameter.

The study is clearly written, logical and easy to follow. The shortcomings are identified and suggestions for prospective studies are stated. 

My only question is why the authors did not use logistic regression to identify potential underlying mechanisms in this dichotomic problem?

Author Response

Thank you very much for your valuable comments

1)The authors in a retrospective chart study investigated fatigue after ischemic and hemorraghic stroke in 178 pts in one Korean institution. 54% of patients were classified as in fatigue, the rest not. Clinical and lab variables were compared between groups (and subgroups), and a few factors proved significant. Linear regression analysis proved sign ificant effects of cognitive decline and depression as well as a biochemical parameter.

The study is clearly written, logical and easy to follow. The shortcomings are identified and suggestions for prospective studies are stated.

My only question is why the authors did not use logistic regression to identify potential underlying mechanisms in this dichotomic problem?

Thank you for your kind and proper pointing out. I concur with your point of view. The dependent variable of regression analysis w as modified to FSS and FAS scores, which are continuous variables, for a more accurate statistical analysis. This, I believe, allowed for more precise statistical analysis to be performed. Thank you once more for your kind remarks.

Reviewer 2 Report

The abstract is too long and you repeated some ideas. It should be a total of about 200 words maximum. The abstract should be a single paragraph and should follow the style of structured abstracts, but without headings as Objectives, Materials and Methods, Results, Conclusions.

A mixture of physical and emotional factors contribute to experiencing fatigue after stroke, being an interesting subject to research. You could discuss it as influencing also the health related quality of life as in the research https://pubmed.ncbi.nlm.nih.gov/34356168/ .

The methods are very well described, but the statistical analysis did not include the assessment of normal distribution of the variables. You can not perform t-test if the continuous variable is not normal distributed, for example. Also, you have a small number of patients with hemorrhagic stroke to use t-test or Chi-square test.

Table 5 must be rewritten to be better understood.

You used FAS and FSS only to group the patients into the two groups. It would be interesting to assess some correlations between the level of fatigue and other indicators, for example, depression or some markers.

Author Response

Thank you very much for your valuable comments:

1) The abstract is too long and you repeated some ideas. It should be a total of about 200 words maximum. The abstract should be a single paragraph and should follow the style of structured abstracts, but without headings as Objectives, Materials and Methods, Results, Conclusions.

Thank you for your kind and proper pointing out. We removed all of abstarct's subheadings, erased extraneous phrases, and rewrote it to be 200 words or less, as you suggested.

2) A mixture of physical and emotional factors contribute to experiencing fatigue after stroke, being an interesting subject to research. You could discuss it as influencing also the health related quality of life as in the research https://pubmed.ncbi.nlm.nih.gov/34356168/ .

Thank you for your kind and proper pointing out. However, we have already written in the first paragraph of Introduction that poststroke fatigue can adversely affect the quality of life of patients. Please understand this.

3) The methods are very well described, but the statistical analysis did not include the assessment of normal distribution of the variables. You can not perform t-test if the continuous variable is not normal distributed, for example. Also, you have a small number of patients with hemorrhagic stroke to use t-test or Chi-square test.

Thank you for your kind and proper pointing out. We agree with your point. We performed a normal distribution test for all variables included in this study, and statistical analysis was performed again using non-parametric statistical methods for variables that did not show a normal distribution.

4) Table 5 must be rewritten to be better understood.

Thank you for your kind and proper pointing out. To make it easier for readers to understand, we've added the heading "Confounding factor" to the first line to better understand why each variable is included.

5) You used FAS and FSS only to group the patients into the two groups. It would be interesting to assess some correlations between the level of fatigue and other indicators, for example, depression or some markers.

Thank you for your kind and proper pointing out. We agree with your point. By synthesizing the opinions of another reviewer, during regression analysis, the dependent variable was changed to FAS and FSS scores, which are continuous variables and suggestive of severity of fatigue, and reanalyzed.

Round 2

Reviewer 2 Report

Table 5 and multivariate analysis must be corrected. In the text, you say only some of the confounding factors you introduced in the table. I propose to show only the values for demonstrating the positive/negative association and the p-value. Why do you put the confounding factors in the table without showing the values you obtained in the univariate analysis? And arrange the table so the variable must be in the same line with its values.
